# Control of cystic echinococcosis in the Middle Atlas, Morocco: Field evaluation of the EG95 vaccine in sheep and cesticide treatment in dogs

Fatimaezzahra Amarir[1,2,3]*, Abdelkbir Rhalem[1], Abderrahim Sadak[3], Marianne Raes[2], Mohamed Oukessou[4], Aouatif Saadi[1,5], Mohammed Bouslikhane[1], Charles G. Gauci[6], Marshall W. Lightowlers[6], Nathalie Kirschvink[2], Tanguy Marcotty[2]

1 Laboratory of Parasitology, Department of Pathology and Veterinary Public Health, Hassan II Agronomy and Veterinary Institute (IAV), Rabat, Morocco, 2 Integrated Veterinary Research Unit, Department of Veterinary Medicine, Namur Research Institute for Life Sciences (NARILIS), University of Namur, Belgium, 3 Laboratory of Zoology and General Biology, Parasitological and Ecological Unit, Faculty of Sciences, Mohammed V University, Rabat, Morocco, 4 Unit of Physiology and Therapeutics, Department of Veterinary Biological and Pharmaceutical Sciences, Hassan II Agronomic and Veterinary Institute, Rabat, Morocco, 5 Fundamental and Applied Research for Animals and Health (FARAH), University of Liège, Liège, Belgium, 6 Faculty of Veterinary and Agricultural Sciences, University of Melbourne, Australia

* amarir.f@gmail.com

**Data Availability Statement:** All relevant data are within the manuscript.

## Abstract

### Background

Cystic echinococcosis (CE) is an important cause of human morbidity and mortality worldwide, particularly in Morocco and other North African countries.

### Methodology/Principal findings

We investigated the potential of three strategies to reduce *Echinococcus granulosus* transmission: (1) 4-monthly treatment of dogs with praziquantel, (2) vaccination of sheep with the EG95 vaccine and (3) a combination of both measures. These measures were implemented during four consecutive years in different areas of the Middle Atlas Mountains in Morocco. The outcome of the interventions was assessed through hydatid cyst (viable and non-viable) counts in liver and lungs using necropsy or *in vivo* ultrasound examination of the liver. A total of 402 lambs were recruited for annual vaccination with the EG95 anti-*E. granulosus* vaccine and 395 similar lambs were selected as non-vaccinated controls.

At approximately four years of age the relative risk (estimated as odds ratio) for vaccinated sheep to have viable hydatid cysts compared with non-vaccinated controls was 3% (9.37% of the vaccinated sheep were found infected while 72.82% of the controls were infected; p = 0.002). The number of viable cysts in vaccinated animals was reduced by approximately 97% (mean counts were 0.28 and 9.18 respectively; p<0.001). An average of 595 owned dogs received 4-monthly treatment during the 44 months trial, corresponding to 91% of the owned dog population. Approximately, 5% of them were examined for *E. granulosus* adult worms by arecoline purge or eggs in feces (confirmed by PCR). The proportion

**Funding:** This study has received funding for the various works carried out in the field, from the Academy of Research and Higher Education (ARES) of Belgium, Namur Research Institut (NARILIS), University of Namur, Belgium, Hassan II Agronomy and Veterinary Institute (IAV), Rabat, Morocco and Australian National Health and Medical Research Council Grant GTN1105448 (MWL). The funders had no role in study design, data collection and analysis, decision to publish, or preparation of the manuscript.

**Competing interests:** The authors have declared that no competing interests exist.

of infected dogs significantly decreased after treatment (12% versus 35%; p<0.001). Post-treatment incidence of re-infestation corresponded to a monthly risk of 4% (95% CI: 3–6%). Treatment of owned dogs on a 4-monthly basis did not reduce the level of transmission of *E. granulosus* to sheep, nor did it enhance the level of control generated by vaccination of sheep with EG95, possibly because of unowned dogs and wild canids were not treated.

## Conclusions/Significance

These data suggest that vaccination of sheep with EG95 has the potential to reduce the level of CE in Morocco and in other parts of the world with similar transmission dynamics. Under the epidemiological circumstances existing in the trial area, 4-monthly treatment of owned dogs with praziquantel was insufficient to have a major impact of *E. granulosus* transmission to sheep.

### Author summary

Cystic echinococcosis remains a major public health problem in Morocco. It is a major zoonosis affecting humans and animals caused by the larval stage of the cestode *Echinococcus granulosus*. The dog (final host), plays an essential role in the dissemination of eggs in the environment via its feces. The rural and poorest regions in the Middle Atlas that practice extensive sheep farming (intermediate host) are severely affected. Women and children are particularly affected by this zoonosis. Despite previous efforts done by the Moroccan authorities to reduce the incidence of infestation, these measures have been insufficient to control the disease. Through our study protocols in natural field conditions, we have shown that vaccination of the intermediate host is an effective control option in the Moroccan context, with an immune protection rate of 97%. Regular chemotherapy (4 months intervals) of owned dogs only proved to have little efficiency on incidence in sheep.

## Introduction

Cystic echinococcosis is one of the most important zoonotic diseases prevailing in different parts of Morocco [1, 2], as well as in other countries of North Africa and the Middle East [3]. Cystic echinococcosis is caused by the larval stage of the cestode *Echinococcus granulosus* [4] in humans as accidental hosts and a range of herbivores as natural intermediate hosts [5]. Dogs act as the definitive hosts and play an essential role in the dissemination of parasite eggs into the environment via their feces and contaminated fur [6–8]. In Morocco, areas with extensive sheep farming are severely affected since sheep are a very common intermediate host. The parasite's cycle is poorly understood by the population, which severely affects compliance to the recommended hygiene measures aiming at reducing disease transmission [9]. Battelli [10] advocated for the integration of social, political and economic factors and argued for better use of available resources and adaptation of control strategies to the regional context.

Although a control strategy was formulated for Morocco in 2007 [11], inconsistent efforts were made to implement the strategy and little seems to have changed in the incidence of cystic echinococcosis in the country in the ensuing years. The strategy aimed to (i) break the biological cycle of the parasite using a cestocidal anthelmintic in dogs and reduce infection in

ruminants and humans, (ii) detect and treat human cases early and (iii) elaborate an appropriate legislative and regulatory arsenal [11]. In other parts of the world, some control strategies have been successful, particularly those undertaken on islands (Iceland, New Zealand, Tasmania, Islas Malvinas and Cyprus), as well as in a limited number of continental regions (Chile, Argentina, Uruguay) [12]. Successful control programs focused on the improved infrastructure of slaughterhouses, the use of purgatives and cestocides in dogs and the regulation of sheep slaughter at home. A number of control programs had limited success [12]. The causes of failure to control cystic echinococcosis include: inadequate management of stray dogs, reliance on dog owners to implement dog treatments, inadequate public funding of control measures, inadequate baseline data collected to measure the progress of the program, the lack of qualified personnel and the premature termination of control program funding [12, 13].

A vaccine against *E. granulosus* was developed for use in sheep and other hosts. The vaccine, known as EG95 described originally by Lightowlers et al. [14] consists of a recombinant protein (16.6 kDa) combined to glutathione-S-transferase (GST) and expressed in *Escherichia coli*. More recently a slight variant of the vaccine, being of 13.4kDa, was described and designated EG95NC⁻ (hereafter referred to as EG95), which has improved productivity in *E. coli* [15]. This vaccine induces an immune response which protects against a subsequent challenge infection with *E. granulosus* eggs [14]. The recombinant antigen is administered to animals together with Quil A or another suitable adjuvant. It typically induces >95% protection against infection while causing minimal injection site reactions or other side effects [14–16]. The vaccine was effective in reducing cystic echinococcosis in sheep in a field trial in Argentina [17] and has been adopted in China as part of the country's Nation Program for the Control of Echinococcosis [18].

Togerson and Heath [19, 20] used mathematical modelling to predict the impact of various options for control of cystic echinococcosis and recommended a combination of vaccination of intermediate hosts together with regular treatment of dogs with praziquantel [19, 20]. These two components were predicted to break the live cycle of the parasite and their synergy predicted to decrease the time needed to achieve cystic echinococcosis control.

A high prevalence of *E. granulosus* infection has been recorded in animals and humans in the Atlas Mountains area of Morocco [21]. This region is a hotspot of hydatid cyst infestation with an infestation prevalence of 91.7% in adult ewes (age > 4 years), and a prevalence of 1.9% in humans [22]. A large canid population is present in this area which includes owned dogs, stray dogs, jackals and foxes [23]. In 2019, the prevalence of *E. granulosus* reached 23 to 39% in owned dogs and 51% to 68% in stray dogs whereas the monthly incidence risk was 2 to 8% and 19 to 41% in owned and stray dogs respectively [6]. In order to evaluate the relevance of control strategies for cystic echinococcosis in Morocco, we carried out a field trial to assess the extent to which vaccination of sheep with the EG95 vaccine from an early age, with or without chemical treatment of *E. granulosus* infection in owned dogs (using praziquantel at four months intervals) could reduce the occurrence of cystic echinococcosis in adult sheep. The study was carried out in the Atlas Mountains in Morocco.

## Methods

### Ethics statement

This work has been authorized by the animal welfare and ethics committee in Hassan II Agronomy and Veterinary Institute (IAV), Rabat, Morocco., in 2015. The protocol was applied according to the international standards cited in many scientific references and in the 2012 OIE Manual titled "Manual of diagnostic tests and vaccines for terrestrial animals".

## Study area

Study sites were selected in the Middle Atlas Mountains, Morocco: El Kbab and Ait Ishaq (about 1100 m of altitude with surface area of 344 Km$^2$ and 373 km$^2$, respectively) in Khenifra province and Ain Leuh (Ain Leuh 1 at 1300m of altitude with surface area of 3.65 Km$^2$ and Ain Leuh 2 at1650 m of altitude with surface area of 5.91 Km$^2$) in Ifrane province (Fig 1). Sites are located between the Rif and the High Atlas, covering a total area of 2.3 million hectares. The Middle Atlas is a sheep breeding area with three million head of the Timahdit breed. The Middle Atlas comprises two very distinct geological structures: table land and fold crossed by rivers, the Oum-Rabiâa and the Oued Guigou. The chain of the Middle Atlas consists of mountainous peaks including Tichoukt (2700 m), Jbel Fazaz, Bou Iblane and Bou Nasser (over 3000 m) which overlook the plain of Moulouya to the south. These mountainous regions are characterized by a continental Mediterranean mountain climate, cold and rainy in winter, hot and dry in summer. Annual precipitations vary between 350 (south-west) and 1100 mm (north-East). Snowfall occurs from December to April with variable accumulation ranging from 20 to 60 cm.

El Kbab, Ait Ishaq and Ain Leuh are rural areas, inhabited by Berbers, a Moroccan community speaking the Amazigh language. Their main source of income is based on farming Timahdit sheep. This breed is known for the production of meat and wool. Farmers slaughter sheep for their own consumption on an ad hoc basis, but particularly on the feast of sacrifice (religious event) and during the sheep-shearing ceremony in the late spring. Male sheep tend to be slaughtered at an age of approximately one year while females, which are most frequently consumed in the rural regions, are slaughtered at an older age, up to six years. Farmers practice extensive and silvopastoral breeding. This practice begins in the morning and ends in the evening. Farmers are accompanied by their dogs, which help them to manage the movements of the sheep. Farmers are accustomed to feeding their dogs the viscera of sheep at slaughter, and those who do not feed it to dogs deliberately, generally do not destroy the infested offal, but discard it into the environment. Dogs are useful to protect the herds against wild predators such as jackals, which are abundant in these regions since their hunting is forbidden by law since 2012.

The choice of these regions for the study was based on the endemicity of cystic echinococcosis [24] and the importance and herd size of the sheep population, estimated to be 240,000 and 65,000 animals in Aïn Leuh and El Kbab/Ait Ishaq regions, respectively [25]. The human population is about 9,600 in Ain Leuh and 35,000 in the site of El Kbab and Ait Ishaq [26] with an annual incidence of cystic echinococcosis in the human population of 16 and 7.24 cases/100 000 inhabitants, respectively [27]. Based on the data published on *E. granulosus* prevalence

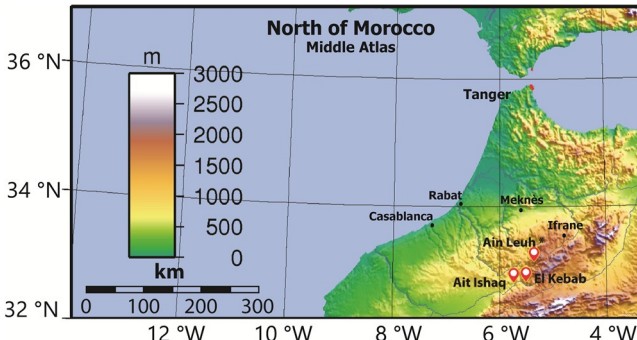

**Fig 1. Location of the three study sites in the Middle Atlas, Morocco (https://fr.wikipedia.org/wiki/G%C3%A9ographie_du_Maroc).**

in the region [6], it was expected that about 50% of control sheep would present detectable infection after four years of natural exposure.

## Study design

A total of 32 sheep breeders were recruited, including 10 in El kbab, 2 in Ait Ishak, 12 in Ain Leuh 1 and 8 in Ain Leuh 2. The field vaccination trial took place in two different settings: one where dogs were regularly treated with a cesticide (Ain Leuh 2) and the other where no treatment was given to dogs (Ain Leuh 1, Ait Ishaq, El Kbab). In each of these two settings, young female lambs were randomly selected at each breeder's farm and allocated randomly to either the vaccination or the control group. Male sheep were not recruited for the trial because they are rarely kept beyond 12 months of age and *E. granulosus* cysts have rarely been found to be fertile in sheep less than 2 years of age [28, 29]. Farmers were recruited based on their agreement to cooperate with the requirements of the study including permission to allow their animals to be vaccinated and monitored. The recruited sheep were followed for about a four years time period during which a natural *E. granulosus* challenge was expected to occur. The infectious status of the sheep was then assessed through antemortem ultrasounds examination or post-mortem dissection, depending on the willingness of their owners. Sheep were slaughtered at the local abattoir with the owners' consent. The numbers of sheep recruited, allocated to the different groups and assessed in each site are shown in Table 1.

A total of 797 female lambs were recruited at the age of one week in October 2015. Among them, 402 lambs were selected for vaccination and 395 animals were used as unvaccinated controls. All animals were identified by ear tag and tattooed with numbers. An important number of animals (vaccinated and controls) were lost from the beginning of the study to the end (among 797 ewes recruited at the beginning of the study, 393 were lost at the end). This was mainly due to animals' sales as result of poor cooperation of some breeders, as well as to sheep mortality.

## Immunization scheme

In December 2015, the vaccinated group received a primary EG95 vaccine immunization at about two months of age subcutaneously in the right neck region. The EG95 vaccine was prepared at the University of Melbourne, Australia, lyophilized together with the Quil A adjuvant and maintained refrigerated until use [15]. Vaccine vials were rehydrated with sterile distilled water on the day they were to be used. The final dose was 1ml containing 50 μg antigen and 1mg adjuvant. A boost vaccine was administered one month later and subsequently annually

**Table 1. Number of sheep recruited, vaccinated and assessed for *E. granulosus* infection by ultrasound (US) or after slaughter in each study site and the number of lost animals (drop outs); US and slaughter numbers are mutually exclusive.**

| Area | Dog chemotherapy | Sheep vaccination | Number recruited | Number. of drop outs | Number US scanned after 4 years of exposure | Number dissected after 4 years of exposure |
|---|---|---|---|---|---|---|
| Ain Leuh 1 | - | + | 150 | 84 | 0 | 66 |
|  | - | - | 149 | 11 | 70 | 68 |
| Ain Leuh 2 | + | + | 80 | 57 | 3 | 20 |
|  | + | - | 81 | 18 | 40 | 23 |
| Ait Ishaq | - | + | 90 | 62 | 18 | 10 |
|  | - | - | 89 | 45 | 34 | 10 |
| El Kbab | - | + | 82 | 71 | 0 | 11 |
|  | - | - | 76 | 45 | 18 | 13 |
| Total |  |  | 797 | 393 | 183 | 221 |

during three years, in February 2016, 2017 and 2018. Only animals which became unavailable due to death or sale were lost to the study prior to its completion. All others received the full complement of vaccinations.

## Cestocidal treatment of dogs and evaluation of *E. granulosus* infection in dogs

In the study site where dogs were treated, all owned dogs were attempted to be treated every four months with praziquantel (5 mg/kg), except puppies <2 months of age, from December 2015 to April 2019. The number of treated dogs (500 to 700 per treatment episode) fluctuated during the study due to mortality, births, transhumance, dog sales or losses (Table 4). The choice of the site (Ain Leuh 2) was based on the farmers' engagement to cooperate and authorization of treatment and monitoring of their dogs. Also, this site was distant enough from other regions to prevent dogs from one study site potentially interfering with *E. granulosus* transmission at other study sites (Ain Leuh 2 is 115 km from Ain Leuh 1). Additional sites were initially planned to carry out chemotherapy in dogs but were abandoned due to logistical problems. In the absence of repetitions, Ain Leuh 1 and Ain Leuh 2 were assumed comparable since they share a number of ecological and epidemiological characteristics: rural areas of the Ifrane province, characterized by the type of sheep breeding (silvopasture) and a large canine population. In Ain Leuh (1 and 2), 3,382 owned dogs were enumerated [30] with 0.35 dogs per inhabitant, which is more than in other regions of the study. At each treatment administrated by the investigators (A FE), faecal samples (non pooled) of 5 to 10% of the treated dogs (all > 18 months of age) were screened for *E. granulosus* adult worms or eggs (including confirmation by PCR according to Amarir et al [6]) after administration of arecoline hydrobromide (Tokyo Chemical Industry) 4 mg/kg body weight *per os* [6].

## Ultrasound screening of sheep

Ultrasound screening was performed at the end of the trial and used as an attempt to reduce the number of sheep slaughtered without affecting the statistical significance of the study. After 44 months of field exposure, 183 sheep were scanned with ultrasound, using a mobile ultrasound machine (TITAN ultrasound System ref P08830-02 SonoSite) with 3.5 MHz sectorial transducer. Sheep were prepared for ultrasound examination by shaving their wool on the right side of the body between the 7th and 12th intercostal spaces according to the protocol described by Hussein and Elrashidy [31]. Only a part of the liver could be examined: the right lobe, around the gallbladder, the sus-hepatic vein and portal vein and only a small part of the left lobe. Lungs could not be scanned because of their echogenicity. The technique allowed the discrimination between liquid-filled and calcified liver cysts, considered viable and dead, respectively.

## Necropsy of slaughtered sheep

In each region, animals to be slaughtered were identified by group and site and taken to the slaughterhouses of each region. At slaughter the liver, lung, spleen, heart and kidneys were examined for the presence of hydatid cysts. The kidneys and spleen were palpated to identify possible cysts, while the liver and lungs were cut into strips about 1 cm thick, then further cut at about 4mm spacing using a scalpel blade, in search of cysts. Again, cysts were classified as viable (filled with liquid) and non-viable. Viable metacestodes of *Taenia hydatigena* were differentiated macroscopically. It was not possible to determine whether non-viable, calcified lesions were due to *E. granulosus*, *T. hydatigena* or some other cause. A total of 221 sheep were examined at post-mortem.

## Statistical analysis

Different statistical models were developed in Stata 11 (Stata Corp) to:

- Evaluate the overall exposure (4 years) of control sheep in the different study sites and the corresponding annual incidence risk.

- Estimate prevalence and incidence risk of *E. granulosus* in dogs in Ain Leu 2 (area where dogs were treated).

- Compare the incidence in sheep in Ain Leu 1 and 2, where dogs were treated or not.

- Quantify the level of protection induced by the vaccine in sheep.

The proportion of non-vaccinated sheep detected with live *E. granulosus* cysts at necropsy and at necropsy or US scan in areas where dogs were not treated was analysed using logistic regressions (model 1 for necropsy data and model 3 for necropsy + US data). Sites and diagnostic tool (ultrasound or necropsy) were used as explanatory variables. Likelihood ratio tests were applied to compare nested models after removal of insignificant explanatory variables. Annual incidence risk was calculated assuming an exposure of 4 years and a period of 1 year needed for the cyst to become detectable after infection:

$$I_a = 1 - (1 - p2I_e)(1/3)$$

$I_a$ = annual incidence risk; $I_e$ = incidence risk for the four years exposure period

Similarly, abundance of cysts at necropsy was analyzed in a negative binomial regression (model 2) using sites as explanatory variable (US scanning did not allow accurate quantification of cysts since only part of the liver was screened).

The proportion of infected dogs in Aïn Leu 2 was analyzed using a logistic regression and time as categorical explanatory variable (model 4). This model was compared using a likelihood ratio test to a logistic regression using before or after the first treatment as single binary explanatory variable. Monthly incidence risk was calculated assuming an exposure of four months and a one month period needed to detect the infection[6] and considering that all tested dogs had been treated four months before:

$$I_m = 1 - (1 - I_e)(1/3).$$

$I_m$ = monthly incidence risk; $I_e$ = incidence risk for the 4-month exposure period

The absence of seasonal effect on incidence in dogs was verified using a likelihood ratio test comparing the latter model with a model including season (April, August or December) as categorical explanatory variable.

Cyst prevalence and abundance in sheep were compared in Ain Leuh 1 and 2 to evaluate the effect of dog therapy. Logistic (model 5 for necropsy data and model 7 for necropsy + US data) and negative binomial (model 6) regressions were applied as above on binary and count data, respectively, using sheep vaccination, dog therapy and diagnosis as binary explanatory variables. Diagnosis was dropped as explanatory variable if p<0.05 in likelihood ratio test.

Finally, prevalence and abundance of viable and dead cysts were compared in vaccinated and control sheep using robust logistic and negative binomial regressions, respectively. Viable and dead cysts were used as responses in separate models (models 8–13, Table 2). Sheep vaccination and diagnostic method were used as binary explanatory variables. The latter was to drop if not significant (p<0.05 in likelihood ratio tests of non-robust regressions). Sites were used as random clusters.

Odds ratios (OR) were obtained from logistic regressions and used as proxy of relative risks whereas incidence rate ratios (IRR) generated by negative binomial regressions are indicative of relative abundance.

## Results

### *E. granulosus* incidence in sheep in the absence of dog therapy and sheep vaccination

The proportion of infected sheep and mean number of viable cysts found in non-vaccinated sheep from areas where dogs did not receive chemotherapy are summarized in Table 3. The prevalence recorded at necropsy was significantly higher in Ain Leuh than in Ait Ishaq and El Kbab (p = 0.001 for both). This difference is significant between Ain Leuh 1 and Ait Ishaq (p = 0.03) when including ultrasound data although there was no significant effect of the diagnostic method (p = 0.74). No variations were detected in cyst abundance at necropsy across sites (likelihood ratio test: p = 0.3).

### The effect of dog therapy on *E. granulosus* prevalence in dogs

Details about the number of dogs treated with praziquantel at 4-month intervals and results of diagnostic tests in dogs are shown in Table 4. Over the duration of the trial, 6545 dog cestocidal treatments were delivered, ranging between 556 and 661 treatments at each treatment time. Arecoline purge assessments were undertaken on a total of 387 dogs, ranging between 27 and 63 at each assessment time. Most samples that were recorded positive for *E. granulosus* contained an abundance of worms, with only five samples positive for taeniid eggs without adult worms being found in the purge.

The proportion of infected dogs was significantly lower after treatment (model 4: OR = 0.26; p<0.001). The variability of the rate of infection observed during the period between the second (April 16) and the last treatment (April 19) was not significant (likelihood ratio test: p = 0.58), nor was the seasonal effect (likelihood ratio test: p = 0.89). At the start of the study (December 15), 35% of the dogs were infected (95% CI: 24–48%) whereas after treatment, 12% were infected (95% CI: 9–16%) on average. Given a treatment every 4 months and an assumed period of 1 month required for the infection to be detectable, this corresponds to a monthly incidence rate of 4% (95% CI: 3–6%).

### The effect of dog therapy on *E. granulosus* incidence in sheep

Cyst incidence and abundance were compared between Ain Leuh 1 (without dog therapy) and Ain Leuh 2 (with dog therapy). Neither incidence nor abundance differed significantly among the two sites, though the effect of vaccination was significant, both on prevalence and abundance (Table 4). Diagnostic tools had no significant effect on the analysis (likelihood ratio test:

**Table 2. Statistical models used to evaluate the effect of sheep vaccination on the occurrence and abundance of live and dead cysts detected in sheep at necropsy and ultrasounds in different sites.**

| Model | Regression | Response |
|---|---|---|
| 8 | Logistic | Binary: sheep with live cysts at necropsy |
| 9 | Logistic | Binary: sheep with dead cysts at necropsy |
| 10 | Negative binomial | Count: number of live cysts in sheep at necropsy |
| 11 | Negative binomial | Count: number of dead cysts in sheep at necropsy |
| 12 | Logistic | Binary: sheep with live cysts at necropsy or US |
| 13 | Logistic | Binary: sheep with dead cysts at necropsy or US |

**Table 3. Proportion of non-vaccinated sheep detected as having viable *E. granulosus* cysts at necropsy alone or by either necropsy or ultrasound examination, calculated annual incidence risk, and mean number of viable cysts detected per animal with 95% confidence intervals.**

| Place | Viable cysts detected at necropsy | | | | Viable cysts detected by necropsy or ultrasound | | |
|---|---|---|---|---|---|---|---|
| | Number of observations | Proportion of infected sheep (model 1) | Annual risk (derived from model 1) | Mean number of viable cysts per infected animal (model 2) | Number of observations | Proportion of infected sheep (model 3) | Annual risk (derived from model 3) |
| Ain Leuh 1 | 68 | 0.88 (0.78–0.94) | 0.51 (0.39–0.6) | 10.35 (7.5–14.28) | 138 | 0.78 (0.7–0.84) | 0.39 (0.33–0.46) |
| Ait Ishaq | 10 | 0.4 (0.17–0.7) | 0.15 (0.05–0.33) | 4.2 (1.75–10.03) | 44 | 0.61 (0.46–0.74) | 0.27 (0.18–0.36) |
| El Kbab | 13 | 0.46 (0.22–0.71) | 0.18 (0.08–0.34) | 8.3 (3.95–17.43) | 31 | 0.74 (0.56–0.86) | 0.36 (0.24–0.48) |

p = 0.2). Dog therapy OR and IRR present large confidence intervals as explanatory variable (Table 5), corresponding to rather imprecise estimations. Yet, the lower limits are larger than the corresponding vaccination OR and IRR denoting a weaker effect of dog therapy compared to vaccination.

The proportions of sheep infected with viable *E. granulosus* cysts and mean number of viable cysts in Ain Leuh 1 and 2 are summarized in Table 6.

## The effect of sheep vaccination on *E. granulosus* occurrence in sheep

Vaccinated sheep had a relative risk of 3% to be infected with viable cysts compared to unvaccinated controls (OR = 0.03 and 0.038 at necropsy and necropsy + US, respectively) and the relative abundance of viable cysts was 3% (Table 7). The prevalence and abundance of dead cysts was higher in vaccinated sheep than non-vaccinated sheep. There was no significant difference in the prevalence and abundance of dead cysts in vaccinated sheep compared to non-vaccinated sheep. The mean proportions of vaccinated and control sheep infected with viable and dead cysts are presented in Table 8 with 95% confidence intervals. Mean abundances are also displayed in Table 8. Diagnosis methods had no significant effect. Confidence intervals were particularly wide for the proportion of animals with viable cysts at necropsy. This was associated to a rather large design effect (DEFT = 2.2; in a robust model, DEFT corresponds to the multiplication factor of the standard error of a non-robust model to account for intra-cluster correlation of the observations), indicating an important variability across sites. This problem was not observed when US data was added (DEFT = 1.01), which generated more precise estimates. Abundance of cysts in sheep were also estimated with a satisfactory level of precision

**Table 4. Number of owned dogs treated and tested for *E. granulosus* infection in Ain Leuh 2 where dogs were treated at 4-month intervals with praziquantel. Initial assessments were undertaken in 2015 and the last assessments in 2019. Testing of praziquantel treated dogs was undertaken at the time of, but prior to, treatment.**

| Date (month/year) | 12/15 | 4/16 | 8/16 | 12/16 | 4/17 | 8/17 | 12/17 | 4/18 | 8/18 | 12/18 | 4/19 |
|---|---|---|---|---|---|---|---|---|---|---|---|
| N. treated dogs (a) | 601 | 556 | 660 | 661 | 535 | 582 | 614 | 572 | 630 | 584 | 550 |
| N. tested dogs (b) | 63 | 29 | 34 | 37 | 27 | 30 | 37 | 32 | 35 | 31 | 32 |
| % tested dog (100*b/a) | 10 | 5 | 5 | 6 | 5 | 5 | 6 | 6 | 6 | 5 | 6 |
| N. dogs with adult worms (c) | 19 | 5 | 2 | 5 | 5 | 3 | 4 | 5 | 2 | 4 | 3 |
| N. dogs positive at PCR* (d) | 3 | 0 | 0 | 0 | 1 | 0 | 0 | 1 | 0 | 0 | 0 |
| N. positive dogs (c+d) | 22 | 5 | 2 | 5 | 6 | 3 | 4 | 6 | 2 | 4 | 3 |
| % positive dogs (100*(c+d)/b)) | 35 | 17 | 6 | 14 | 22 | 10 | 11 | 19 | 6 | 13 | 9 |
| Consolidated % positive dogs before and after first treatment | 35 | 12.7 | | | | | | | | | |

* Dogs without worms but with taeniid eggs which were confirmed as positive for *E. granulosus* by PCR. Data before first treatment is in italic.

**Table 5. Effect of dog therapy on live cyst prevalence and abundance in sheep: odds ratios (for binary responses) or incidence rate ratios (for count response) and p values obtained for the two explanatory variables assessed in three models in Ain Leuh 1 and 2.**

| Model | Response | Explanatory variable | OR/IRR | P |
|---|---|---|---|---|
| 5 | Prevalence at necropsy | Vaccination* | 0.01 (0.004–0.04) | <0.001 |
| | | Dog therapy | 0.35 (0.12–1.01) | 0.05 |
| 6 | Abundance at necropsy | Vaccination* | 0.02 (0.01–0.04) | <0.001 |
| | | Dog therapy | 0.70 (0.39–1.26) | 0.23 |
| 7 | Prevalence at necropsy + US | Vaccination* | 0.03 (0.01–0.06) | <0.001 |
| | | Dog therapy | 0.65 (0.34–1.21) | 0.18 |

* The effect of vaccination presented here should not be regarded as a result as such (because it is evaluated more accurately below in different sites) but as a reference to compare the effect of dog therapy.

(DEFT = 0.66 for the vaccination variable and 1.78 for the negative binomial distribution dispersion parameter) though only necropsy data were used to quantify cysts.

## Discussion

A control trial to reduce *E. granulosus* transmission was carried out for the first time under field conditions in Morocco. The choice of the Middle Atlas region for the trial was made because of the importance of sheep breeding, presence of numerous dogs (1.08 dog/person) and high level of *E. granulosus* infection in dogs confirmed recently in the same region [6]. These conditions were considered favourable for field testing of cystic echinococcosis control measures such as sheep vaccination and dog chemotherapy. In our field trial the proportion of sheep in the control group (no vaccination, no treatment of dogs) that were found to be infected with viable hydatid cysts at necropsy after a 4-year exposure was very high, with a significantly higher prevalence observed in Ain Leuh (88%) than in other sites (Ait Ishaq and El Kbab, 40% and 46% respectively; Table 2; p = 0.03). These values correspond to annual incidences rate of 51% in Ain Leuh 1 (95% Cl: 0.39–0.6), 17% in Ait Ishaq (95% Cl: 0.06–0.36) and 18% in El Kbab (95% Cl: 0.08–0.34). Similar results were observed when animals assessed by ultrasound were added to the necropsies, with a high annual incidence rate recorded in Ain Leuh 1 (39% Cl: 0.33–0.46) while there was no significant difference seen in the abundance of live cysts in animals, across the different sites (p = 0.3). These results in control sheep confirmed that an exposure of 4 years was sufficient for many of the animals to become infected. It is considered that a minimum of six months is required for hydatid cysts to appear, but it can take more than a year for them to reach a size easily detectable at necropsy [32]. Taking this into account, the annual incidence rate of infection was calculated on the basis of a 3-year period. A similarly high prevalence of cystic echinococcosis was described recently in the province of Sidi Kacem (64%) indicating that *E. granulosus* transmission is widespread in Morocco [2].

Prior to the implementation of control measures we found a prevalence of *E. granulosus* infection of 35% in owned dogs in Ain Leuh 2 (Table 4). Following treatment, the monthly infection incidence rate in dogs was estimated at 4% (95% CI: 3–6%), assuming a period of one month before a new infection could be detected. This was close to what was estimated in a close vicinity (Had Oued Ifrane) [6]. Our data on the levels of infection with *E. granulosus* found in sheep and in dogs correlated with the high incidence rate of cystic echinococcosis found in the human population in the Middle Atlas. A recent study found cystic echinococcosis infection levels to be as high as 2.6% in some communities when determined by screening patients using abdominal ultrasound [22]. Current estimations of the prevalence of *E.*

**Table 6. Ain Leuh 1 and 2: proportion of sheep with viable *E. granulosus* cysts at necropsy alone or either at necropsy or ultrasound examination and the mean number of viable cysts in vaccinated and control sheep in areas where dogs were treated or not (with 95% confidence intervals).**

| Place | Control procedures | | Viable cysts detected at necropsy | | | Viable cysts detected by necropsy or US | |
|-------|----------------|-----------------|------------------------|------------------------------------|------------------------------------------|------------------------|------------------------------------|
| | Dog therapy | Sheep vaccination | Number of observations | Proportion infected sheep (model 5) | Mean number of viable cysts per animal (model 6) | Number of observations | Proportion infected sheep (model 7) |
| Ain Leuh 1 | - | - | 68 | 0.87 (0.77–0.93) | 10.61 (7.65–14.72) | 138 | 0.77 (0.70–0.83) |
| | - | + | 66 | 0.08 (0.03–0.17) | 0.22 (0.12–0.38) | 66 | 0.08 (0.04–0.17) |
| Ain Leuh 2 | + | - | 23 | 0.71 (0.51–0.85) | 7.40 (4.40–12.45) | 63 | 0.69 (0.57–0.79) |
| | + | + | 20 | 0.03 (0.008–0.09) | 0.15 (0.07–0.32) | 23 | 0.05 (0.02–0.13) |

*granulosus* infections in Morocco are higher than those published [21], suggesting that the disease has progressed gradually. This change may be associated with a prohibition on the hunting of jackals or killing of dogs that was enacted in Morocco in 2012.

The potential protective effect of sheep vaccination on cystic echinococcosis infections was evaluated in vaccinated and non-vaccinated control ewes where the proportion of animals with viable cysts was determined as well as the mean number of viable cysts per animal in those that were assessed at necropsy (Tables 7 and 8). The proportion of sheep with viable hydatid cysts at necropsy was lower in vaccinated sheep (10%) than in control sheep (75%; p = 0.032) representing a 97% (95% CI: 42–99.8%) reduction of the risk of infection among vaccinated ewes (1 –OR). This rather imprecise estimation was due to a relative low number of observations at necropsy (107 vaccinated and 114 control ewes) and the high variability across sites. Our investigations of *E. granulosus* infection levels in dogs in one of the trial sites indicate that the lambs were born into a highly contaminated environment, providing opportunity for exposure to infection early in life. Including ultrasound data generated a more precise protection rate (96%; 95% CI: 89–99%) (Fig 2A). Ultrasound being less sensitive (lungs and parts of the liver cannot be screened) and increasing the number of observations may explain this difference. Indeed, a high level of protection of the animals was also evident from the average number of viable hydatid cysts found in infected sheep among the 221 animals that were evaluated for infection at necropsy. Vaccinated animals had an average burden of viable cysts of 0.28, while in control animals the average burden was 9.10 (p<0.001), representing a reduction in the average burden of 97%. This means that vaccinated animals present far less viable cysts and, consequently, are less likely to be found positive at US examination than at necropsy (Fig 2C).

Data is available from a small number of other field studies of the EG95 vaccine trial. Laurieu et al. published results of a control program undertaken in the Rio Negro province of Argentina lasting more than 8 years [17, 33]. Difficulties with accesses to the control area prevented Larrieu and his colleagues from delivering the vaccination program to a high

**Table 7. Occurrence of viable and dead *E. granulosus* cysts in vaccinated and control sheep: odds ratios (for binary responses) or incidence rate ratios (for count response) and p values obtained for the two response variables (live or dead cyst) assessed in six models.** Explanatory variable is vaccination (reference = control).

| Model | Response | Cyst | OR/IRR (95% CI) | P |
|-------|----------|------|-----------------|---|
| 8 9 | Necropsy incidence | Live | 0.03 (0.002–0.52) | 0.029 |
| | | Dead | 1.22 (0.70–2.13) | 0.32 |
| 10 11 | Necropsy abundance | Live | 0.03 (0.017–0.052) | 0.001 |
| | | Dead | 1.45 (1.01–2.05) | 0.046 |
| 12 13 | Necropsy + US incidence | Live | 0.038 (0.01–0.11) | 0.002 |
| | | Dead | 1.35 (1.17–1.56) | 0.07 |

**Table 8. Prevalence and abundance of viable and dead *E. granulosus* cysts detected in vaccinated and control sheep at necropsy and combination of US and necropsy (with 95% CI).**

| Response: Type of cyst | Explanatory variable: Sheep group | Cysts detected at necropsy | | | Cysts detected at necropsy or US | |
|---|---|---|---|---|---|---|
| | | Number of observations | Proportion of infected sheep (models 8 & 9) | Mean number of cysts per animal (models 10 & 11) | Number of observations | Proportion of infected sheep (models 12 & 13) |
| Viable | Control | 114 | 0.75 (0.51–0.90) | 9.10 (7.27–11.37) | 276 | 0.73 (0.64–0.80) |
| | Vaccinated | 107 | 0.10 (0.05–0.18) | 0.28 (0.17–0.45) | 128 | 0.09 (0.07–0.13) |
| Dead | Control | 114 | 0.39 (0.33–0.47) | 1.24 (1.17–1.33) | 276 | 0.38 (0.34–0.42) |
| | Vaccinated | 107 | 0.45 (0.40–0.50) | 1.81 (1.55–2.12) | 128 | 0.45 (0.41–0.49) |

C = control animals or area; V = vaccinated animals; D = dog-treated area

proportion of animals, nevertheless, a reduction in the occurrence of infection and the burden of cystic echinococcosis infection in sheep was achieved (from an initial prevalence before vaccination of 56.3% to 21.1% prevalence after vaccination). The benefits that we found from vaccination with EG95 in Morocco were greater than those reported in Argentina by Larrieu and colleagues; however, there were differences in the programs undertaken in the two areas. In

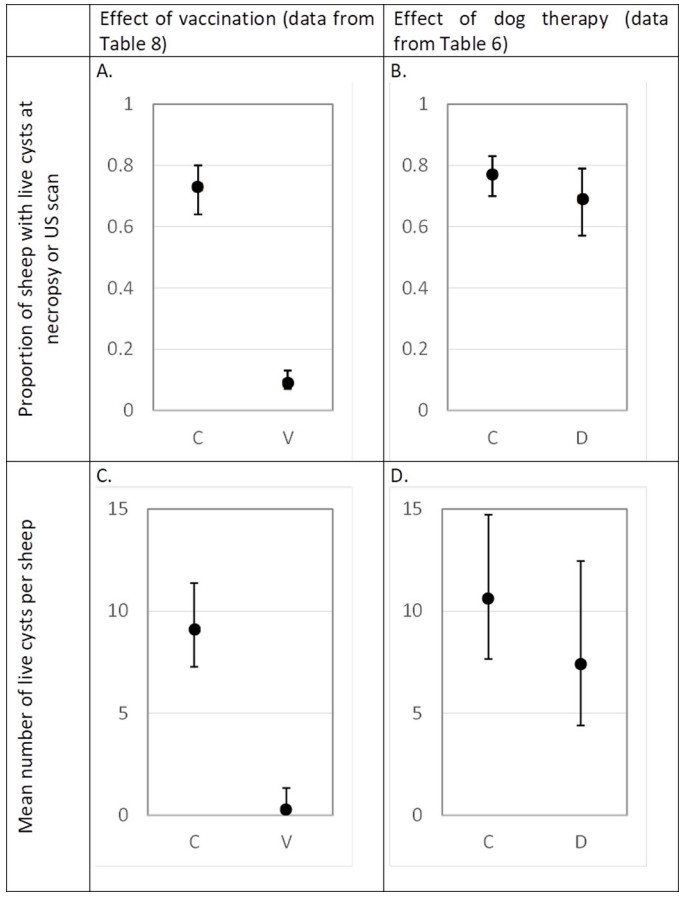

**Fig 2.** Effect of vaccination and dog therapy on prevalence and abundance of live cysts in sheep exposed for 4 years to a natural *E. granulosus* challenge (data extracted from Tables 6 and 8) A: Prevalence of infection in vaccinated and control sheep, B: Prevalence of infection in control sheep with or without treatment of dogs, C: Mean number of live cysts per sheep in vaccinated and control sheep, D: Mean number of live cysts per control sheep with or without treatment of dogs.

our study, selected sheep received two immunizations as lambs followed by annual booster injections during 3 years and the effect of vaccination was assessed by comparing incidence and abundance of live cysts in vaccinated and control animals. In Argentina the sheep received two immunizations as lambs but only a single booster vaccination at one year of age before being assessed for hydatid cysts at the age of five years, whereas we provided additional annual immunizations in our study. Larrieu and colleagues were able to deliver their full vaccination program to only about half the animals, but assessed infection levels in all animals, irrespective [17]. This seems likely to have been a major contributor to the difference seen in the outcomes between the trial in Argentina and ours in Morocco. Heath et al. reviewed an extensive vaccination program undertaken in China [16], however details of the project have not been fully published. In 2017 the Chinese government implemented compulsory vaccination of sheep as part of a national echinococcosis control program [18], however an assessment of the impact of vaccination has yet to be published.

The proportion of sheep with non-viable hydatid cysts did not differ significantly between control (38–39%) and vaccinated sheep (45%). Vaccinated animals that were found to be infected had an average of 1.81 non-viable hydatid cysts whereas in the non-vaccinated animals there was an average of 1.24 non-viable cysts per animal with borderline statistical significance (p = 0.046) (Tables 6 and 7). The reasons for the non-viable lesions found at necropsy in the livers (especially) and lungs remain unknown. There may be fibrotic and calcified lesions remaining after the death of *E. granulosus* parasites, however liver lesions may also be due to the death of migrating *T. hydatigena* metacestodes, or other causes. If caused by *E. granulosus*, their abundance (about 1.5 cysts/animal) was low compared to live cysts in control animals (9 cysts/animal) and, in any case, non-viable cysts do not play any role in the transmission of *E. granulosus* to dogs.

The EG95 vaccine targets a protein which is expressed in eggs and oncospheres but not in established cysts [34] suggesting that it may only affect the parasite early in its development. In most controlled experiments using the EG95 vaccine, vaccinated animals are not always totally protected against a challenge infection with eggs and those parasites that do survive in vaccinated animals are viable [35]. Hence the relatively small number of cysts, viable or not, found in a small proportion of vaccinated sheep in Morocco is consistent with the efficacy of the EG95 vaccine described in controlled experimental trials.

Our trial in Morocco evaluated the impact of 4-monthly chemotherapy in dogs with praziquantel, either as a stand-alone measure or in combination with vaccination of sheep. In the selected vicinity, we attempted to treat all owned dogs every 4 months with praziquantel, i.e. about 600. As indicated above, the dogs of the selected area were highly exposed presenting a prevalence before treatment of 35% and an estimated monthly incidence risk of 4%. Such an incidence causes a detectable prevalence in dogs 4 month after treatment of 12%, significantly lower than the initial prevalence of 35%. Although praziquantel is highly efficient against *E. granulosus*, the effects of the drug in dogs do not persist beyond the day of treatment because the drug is very rapidly eliminated from the circulation and tissues [36, 37]. Consequently, it can reasonably be assumed that all treated dogs were clean after treatment but were subsequently exposed to a high incidence (4% per month), making an average prevalence of about 6% when treated every 4 months (roughly, 0% after treatment and 12% four months later make a mean = (0+12)/2 = 6%). In the area where dogs received 4-monthly chemotherapy, the proportion of non-vaccinated animals found to have viable *E. granulosus* cysts at necropsy was 71% (Table 5). This was less than the proportion of non-vaccinated sheep from the selected control area where the dogs were untreated (87%; p = 0.05; Table 5). The effect of dog therapy was less pronounced in term of infection in sheep when cyst abundance was considered or when ultrasound data were included (Table 5 and Fig 2B and 2D). These estimations lack

precision and may lack accuracy. Indeed, evaluating the effect of dog therapy by comparing only two areas (Ain Leuh 1 and 2) that are a priori similar, but distinct, may generate a bias since the variability observed may be caused by confounding factors. This problem was anticipated but dog therapy and testing proved difficult to implement in all selected places due to lack of collaboration of dog owners. This also impacted the number of observations and, as a consequence, the precision of our estimations.

Among the group of vaccinated sheep from areas where dogs also received praziquantel treatment (Ain Leuh 2), the proportions of sheep infected with viable hydatid cysts were 3% in vaccinated animals compared with 71% in control sheep. The mean number of viable cysts detected per animal was higher in control sheep than in vaccinated animals (Tables 5 and 6). Statistical comparison of the impact of 4-monthly praziquantel treatment together with vaccination of sheep showed in our study area that there was no significant difference indicating that under the transmission conditions pertaining to the control area, 4-monthly praziquantel treatment in owned dogs did not enhance the effect of sheep vaccination over use of the vaccine alone.

Torgerson (2003) used mathematical modelling to simulate control options for cystic echinococcosis [20]. The model predicted that a combination of treatment of dogs and vaccination would synergize to improve the effectiveness of a control program resulting in a greater and faster reduction of cystic echinococcosis in sheep. Torgerson's modelling was based on 6-monthly treatment of 80% of dogs. In our trial we did not obtain a substantial effect on *E. granulosus* transmission of 4-monthly treatment of dogs with praziquantel, nor a synergistic effect of a combination of dog treatment with sheep vaccination. Although we achieved close to 100% treatment of owned dogs in our control area, the incidence rate could be too high or the treatment frequency too low so that a sufficient number of dogs are promptly reinfected and disseminate eggs in the environment. Unfortunately, incidence data in dogs from study sites where dogs were not treated was not available for evaluating the actual effect of chemotherapy on *E. granulosus* prevalence in dogs. Due to logistical constraints, four-monthly dog chemotherapy was not repeated in other sites. Hence, our observations should be viewed cautiously as confounding factors were not controlled.

In addition, there were an unquantified number on non-owned dogs present in the trial region, which were not treated. A study conducted in the Middle Atlas region of Morocco found a very high prevalence of infestation in non-owned dogs, ranging from 51.3% to 68.5% and representing a significantly higher risk of infestation (OR = 14) than in owned dogs [6]. Furthermore, in the trial area and other parts of Morocco there are numerous wild canid species [23] which are other uncontrolled definitive hosts for *E. granulosus* [38–40].

Therefore, our observations do not contradict the model of Torgerson (2003) but illustrate how cumbersome comprehensive dog chemotherapy could be in the field and the level of control that would be required to be effective (frequency of treatment, coverage of owned, roaming and wild canids). Treating regularly pet dogs remains advisable to reduce the risk of transmission to humans. Indeed, deworming of the dog protects man against infestation through licking, caresses and contamination of the shared environment [6].

In the rural conditions of Morocco in which we conducted our study and where *E. granulosus* transmission rates are high, vaccination of sheep proved to be a highly effective tool to assist in the fight against this zoonosis. However, reliance on vaccination alone would require an extensive coverage and a control program to be undertaken for many years until the total renewal of the susceptible livestock population (because sheep that are infected before they are vaccinated remain potentially carriers until they die). To maximize the vaccination coverage, the inclusion of the EG95 in the enterotoxemia vaccine would be promising. According to our field investigation, we estimate that between 60 and 85% of the farmers in the trial areas use an

enterotoxemia vaccine each year in their lambs. Another complementary and valuable control option would be to equip each slaughterhouse, which represent a hotspot for dog infection through consumption of infected organs, with a baking system, allowing to inactivate viable cysts in infected organs by boiling before they are left to carnivore animals [41]. According to Li et al. (2014), boiling livers and lungs, which contain most of the hydatid cysts, could be a simple, effective and inexpensive method in terms of time and energy, to destroy the infective material [42].

In conclusion, we found that under the epidemiological conditions in the Middle Atlas Mountains vaccination of sheep with the EG95 vaccine was highly effective in reducing *E. granulosus* transmission to sheep, whereas 4-monthly treatment of owned dogs with praziquantel was relatively ineffective. We were unable to demonstrate a synergistic effect of vaccination plus 4-monthly dog treatments. This was the first field evaluation of sheep vaccination for the control of cystic echinococcosis in North Africa and will help to inform efforts to control the disease in the region. A bivalent vaccine combining the EG95 vaccine with the enterotoxaemia vaccine will have a definite impact as it remains reasonably cost-acceptable. On the other hand, a major effort to sensitize and communicate with breeders is necessary before implementation. There is an urgent need for enhanced efforts to reduce the transmission of *E. granulosus* in Morocco and other North African countries so as to reduce the burden of cystic echinococcosis in the human population.

## Acknowledgments

Special thanks to Intissar Boukhari, Larbi, Said Suilahi and the entire team of the Department of Pathology and Veterinary Public Health, parasitology lab. Thanks also to all the farmers of the Middle Atlas who participated in this study as well as the local authorities, agents and technicians of the ONSSA and ANOC who brought their help in the field and without forgetting all the people of the Ministry of Health.

## Author Contributions

**Conceptualization:** Abdelkbir Rhalem, Marianne Raes, Marshall W. Lightowlers, Tanguy Marcotty.

**Data curation:** Fatimaezzahra Amarir.

**Formal analysis:** Tanguy Marcotty.

**Funding acquisition:** Abdelkbir Rhalem, Nathalie Kirschvink.

**Investigation:** Fatimaezzahra Amarir, Abdelkbir Rhalem, Aouatif Saadi.

**Methodology:** Fatimaezzahra Amarir, Abdelkbir Rhalem, Marshall W. Lightowlers, Tanguy Marcotty.

**Project administration:** Abdelkbir Rhalem, Marianne Raes, Nathalie Kirschvink.

**Resources:** Mohamed Oukessou, Charles G. Gauci.

**Supervision:** Abdelkbir Rhalem, Abderrahim Sadak, Nathalie Kirschvink.

**Writing – original draft:** Fatimaezzahra Amarir, Marshall W. Lightowlers, Nathalie Kirschvink, Tanguy Marcotty.

**Writing – review & editing:** Fatimaezzahra Amarir, Mohammed Bouslikhane, Marshall W. Lightowlers, Nathalie Kirschvink, Tanguy Marcotty.

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
