## [Decision Letter · Decision Letter 0]

12 Nov 2020

Dear Fatimaezzahra Amarir,

Thank you very much for submitting your manuscript "Control of cystic echinococcosis in the Middle Atlas, Morocco: field evaluation of the EG95 vaccine in sheep and cesticide treatment in dogs" for consideration at PLOS Neglected Tropical Diseases. As with all papers reviewed by the journal, your manuscript was reviewed by members of the editorial board and by several independent reviewers. In light of the reviews (below this email), we would like to invite the resubmission of a significantly-revised version that takes into account the reviewers' comments. 

Please notice that Reviewer 2 has provided a file with comments and further analyses: moroco-plos-review.pdf. Please refer to this file and all the comments in the text below when preparing your responses.

We cannot make any decision about publication until we have seen the revised manuscript and your response to the reviewers' comments. Your revised manuscript is also likely to be sent to reviewers for further evaluation.

Sincerely,

Prof. María-Gloria Basáñez, PhD, MSc

Associate Editor

Pikka Jokelainen

Deputy Editor

Reviewer's Responses to Questions

**Key Review Criteria Required for Acceptance?**

**Methods**

-Are the objectives of the study clearly articulated with a clear testable hypothesis stated?

-Is the study design appropriate to address the stated objectives?

-Is the population clearly described and appropriate for the hypothesis being tested?

-Is the sample size sufficient to ensure adequate power to address the hypothesis being tested?

-Were correct statistical analysis used to support conclusions?

-Are there concerns about ethical or regulatory requirements being met?

Reviewer #1: the objectives are clear, the design is correct but it can be presented more clearly. 

The sample size is acceptable for the study while the statistical analyses are excellent.

Reviewer #2: Mainly, but the authors did not examine untreated dogs as controls, so the hypothesis that there is a synergistic effect of dog treatment and sheep vaccination can not be answered.

**Results**

-Does the analysis presented match the analysis plan?

-Are the results clearly and completely presented?

-Are the figures (Tables, Images) of sufficient quality for clarity?

Reviewer #1: The way of presenting the results can be greatly improved, including tables.

Reviewer #2: Yes, except the dog data and analysis. See enclosed reviewer report.

**Conclusions**

-Are the conclusions supported by the data presented?

-Are the limitations of analysis clearly described?

-Do the authors discuss how these data can be helpful to advance our understanding of the topic under study?

-Is public health relevance addressed?

Reviewer #1: The conclusions are based on the findings but some points can be re-discussed.

Reviewer #2: The conclusion that 4 monthly dog treatment is not synergistic with sheep vaccination can not be supported given the data and analysis reported in the manuscript. Otherwise the efficacy of the sheep vaccination is convincing.

**Editorial and Data Presentation Modifications?**

Reviewer #1: (No Response)

Reviewer #2: (No Response)

**Summary and General Comments**

Reviewer #1: the work presents an experience of CE control in Morocco with PZQ and EG95.

There are very few published experiences of this type, so it is a valuable scientific contribution.

It is a very good paper but it could be improved to make the experience more comprehensive.

It would be very important in the description of the work area to include information on sheep management. For example, what percentage of males are in the flock, are they wool or meat producers? What age of sheep are slaughtered for their own consumption? lambs or old females or old males? is it customary to give the viscera to the dogs of these animals? How many animals does each producer have? Without this information it is difficult to analyze the results and discuss them.

Please, provide further detail for the study design. How were the sheep selected? Are they from a single sheep farmer or from several ones? Were the sheep farmers also randomly selected? In each farm, were there sheep not vaccinated? Can the proportion be estimated of animals that entered the study in relation to the total number of dogs and sheep in the work areas?

It would probably be better to add all the information on the number of existing dogs, number of farms, number of sheep, estimated square Km for each work area in "work area", even with a summary table, including Dog chemotherapy, Sheep vaccination and Number recruited of the current table 1.

Clarify if diagnoses with US were made at the beginning of the experience or only at the end.

How the dogs to be studied were selected by the arecoline test? Were they evaluated in each farm or were pools made? what PCR test was used?

How were the sheep selected by US or necropsy?

Sheep with US and sheep euthanized: are the two diagnostic systems given the same value? I would like to see the different prevalences found within each of these methods.

Table 3. Does the number of observations of Viable cysts detected by necropsy or ultrasound include Viable cysts detected at necropsy?. I don't understand the differences. I would like to see US and necropsy data separately and add a simple estimate of the prevalence of positive animals for each type of technique in the text (number of sheep, number of positive, 95%CI). The same applies to Table 6.

Title 1: Viable cysts detected at necropsy subtitle: Proportion infected sheep and Mean number of viable cysts per animal= Title is confusing.

Table 4. The number of dogs tested 12/5 is pre or post treatment ? I understand is pre-treatment, but the table is not clear; after treatment, “12% were infected (95% CI: 9-16%) on average” this number is not in the table.

Discussion. Include in the comparative analysis between the 2 trials not only the prevalence. Include the number of cysts and their fertility per animal.

Clarify whether the dogs were dewormed by the researchers or by the dog owners or in the presence of the researchers. Try to explain why high prevalences were maintained in dogs. What is the frequency with which a farmer slaughters animals that can infect dogs? (raised for work area)

I would not be so sure that the destruction of the viscera by the farmers works. It never worked in my experience;

“4-monthly treatment of owned dogs with praziquantel was relatively ineffective. We were unable to demonstrate a synergistic effect of vaccination plus 4-monthly dog treatments”…. for the sheep infections. But, for the children ? the remaining infected dogs are enough to infect the sheep. But the decrease in the number of infected dogs may be enough to decrease the infection in children. Consider that CE is a public health problem and not of sheep.

A final summary table could help, including place, control procedure, final proportion of infected sheep, final proportion of infected dogs, with all experiences and the control group.

Reviewer #2: (No Response)

PLOS authors have the option to publish the peer review history of their article (what does this mean?). If published, this will include your full peer review and any attached files.

Reviewer #1: Yes: Edmundo Juan Larrieu

Reviewer #2: Yes: Paul R Torgerson
---

## [Decision Letter · Decision Letter 1]

18 Feb 2021

Dear Mrs., Amarir,

We are pleased to inform you that your manuscript 'Control of cystic echinococcosis in the Middle Atlas, Morocco: field evaluation of the EG95 vaccine in sheep and cesticide treatment in dogs' has been provisionally accepted for publication in PLOS Neglected Tropical Diseases.

Best regards,

Prof. María-Gloria Basáñez, PhD, MSc

Associate Editor

Pikka Jokelainen

Deputy Editor

Reviewer's Responses to Questions

**Key Review Criteria Required for Acceptance?**

**Methods**

-Are the objectives of the study clearly articulated with a clear testable hypothesis stated?

-Is the study design appropriate to address the stated objectives?

-Is the population clearly described and appropriate for the hypothesis being tested?

-Is the sample size sufficient to ensure adequate power to address the hypothesis being tested?

-Were correct statistical analysis used to support conclusions?

-Are there concerns about ethical or regulatory requirements being met?

Reviewer #1: yes

Reviewer #2: Yes

**Results**

-Does the analysis presented match the analysis plan?

-Are the results clearly and completely presented?

-Are the figures (Tables, Images) of sufficient quality for clarity?

Reviewer #1: yes

Reviewer #2: Yes

**Conclusions**

-Are the conclusions supported by the data presented?

-Are the limitations of analysis clearly described?

-Do the authors discuss how these data can be helpful to advance our understanding of the topic under study?

-Is public health relevance addressed?

Reviewer #1: yes

Reviewer #2: Yes

**Editorial and Data Presentation Modifications?**

Reviewer #1: Please add o change Larrieu E, Herrero E, Mujica G, Labanchi JL, Araya D, Grizmado C, et al. Pilot field trial of the EG95

597 vaccine against ovine cystic echinococcosis in Rio Negro, Argentina: Early impact and preliminary

598 data. Acta Trop. 2013;127: 143–151. doi:10.1016/j.actatropica.2013.04.009 by Pilot field trial of the EG95 vaccine against ovine cystic echinococcosis in Rio Negro, Argentina: 8 years of work. Larrieu E, Mujica G, Araya D, Labanchi JL, Arezo M, Herrero E, Santillán G, Vizcaychipi K, Uchiumi L, Salvitti JC, Grizmado C, Calabro A, Talmon G, Sepulveda L, Galvan JM, Cabrera M, Seleiman M, Crowley P, Cespedes G, García Cachau M, Gino L, Molina L, Daffner J, Gauci CG, Donadeu M, Lightowlers MW. Acta Trop. 2019 Mar;191:1-7. doi: 10.1016/j.actatropica.2018.12.025. Epub 2018 Dec 18.

PMID: 30576624

Reviewer #2: (No Response)

**Summary and General Comments**

Reviewer #1: This is an important contribution to the control of CE

Reviewer #2: (No Response)

PLOS authors have the option to publish the peer review history of their article (what does this mean?). If published, this will include your full peer review and any attached files.

Reviewer #1: **Yes: **Edmundo Larrieu

Reviewer #2: **Yes: **Paul R Torgerson

---

## [Editor Report · Acceptance letter]

3 Mar 2021

Dear Mrs., Amarir,

We are delighted to inform you that your manuscript, "Control of cystic echinococcosis in the Middle Atlas, Morocco: field evaluation of the EG95 vaccine in sheep and cesticide treatment in dogs," has been formally accepted for publication in PLOS Neglected Tropical Diseases.

Best regards,

Shaden Kamhawi

co-Editor-in-Chief

Paul Brindley

co-Editor-in-Chief
